# Muon-induced background in a next-generation dark matter experiment based on liquid xenon

V. Pěč[1⋆] and V. A. Kudryavtsev[2]

**1** FZU – Institute of Physics of the Czech Academy of Sciences, Prague, Czech Republic[i)]

**2** Department of Physics and Astronomy, University of Sheffield, Sheffield, United Kingdom

⋆ viktor.pec@fzu.cz

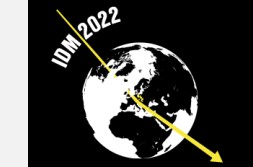

## Abstract

**We have investigated the implication of laboratory depth on the muon induced background in a future dark matter experiment capable of reaching the so-called neutrino floor (fog). Our simulation study focuses on a xenon-based detector with 71 tonnes of active mass, surrounded by additional veto systems including an instrumented water shield. Two locations at the Boulby Underground Laboratory (UK) at depths of 2850 m w. e. and 3575 m w. e. served as a case study. Our results show that less than one event of cosmogenic background is likely to survive standard analysis cuts for 10 years of operation at either location. The largest background component that we identified comes from delayed neutron emission from $^{17}$N which is produced from $^{19}$F in the fluoropolymer components of the experiment. Our results confirm that a dark matter search with sensitivity to the neutrino floor (fog) is viable (from the point of view of cosmogenic backgrounds) in underground laboratories at these levels of rock overburden.**

## 1  Introduction

The choice of underground laboratory for a future high-sensitivity experiment for rare-event searches will be heavily dependent on the depth of the site, as events triggered by cosmic-ray muons can constitute significant background to the signal events. For example, in searches for dark matter (DM) Weakly Interacting Massive Particles (WIMP), isolated neutrons originating in the muon activity can mimic dark matter interaction if scattering only once in the active region producing a single nuclear recoil (NR). We focused our attention on a potential next-generation DM experiment based on Liquid Xenon Time Projection Chamber (LXe-TPC) technology, and we investigated whether the depth of about 3 km w. e. was sufficient for such experiment.

The work was performed as a part of feasibility study for Boulby Mine (UK) to host the next-generation LXe-TPC experiment [1] and we wanted to learn whether the current underground laboratory in Boulby has sufficient depth with respect to the cosmic-ray induced backgrounds. A new larger site would have to be excavated nearby the existing laboratory which is at 1100 m level

---

[i)]Previously at the University of Sheffield

36  underground, or 2850 m w. e., and it is located within a layer of salt (NaCl). We also considered
37  a new site at a deeper location at a 1400 m level, or 3575 m w. e. underground. The Boulby mine
38  served as a case model, however, the results are relevant to sites of similar depth.
39      We have also compared our Geant4 results on neutron production in simplified-geometry sim-
40  ulations for various materials and muon energies with other simulations and experimental data.
41  Some results and distributions of interest are included in Ref. [2].

## 2  Simulation of cosmic-ray muons

### 2.1  Geometry

44  We carried out simulations to determine the rate of potential background events caused by cosmic-
45  ray muons in a next generation dark matter experiment with design based on the design of LUX-
46  ZEPLIN (LZ) detector [3]. The main detector is a dual-phase xenon time projection chamber
47  (hereafter LXe-TPC) containing 70 tonnes of active liquid xenon (LXe), corresponding to a $\sim$10-
48  fold upscale of existing experiments LZ [4] and XENONnT [5].
49      We implemented a simplified detector geometry model with the main elements included: a
50  vacuum cryostat approximately 4 m in diameter and 5 m in height containing the xenon detector,
51  with an anti-coincidence veto system made of gadolinium-loaded liquid scintillator (GdLS), 50 cm
52  thick, surrounding the cryostat, and all submerged in a water tank with 12 m in diameter and 11 m
53  in height for shielding from local radioactivity backgrounds. The water tank was placed into a
54  cylindrical cavern of diameter and height of 30 m. The cavern was surrounded by rock material
55  with at least 5 m on the sides and 7 m and 3 m at the top and the bottom, respectively. We tested
56  two different rock compositions, one with salt (NaCl, 2.17 g/cm$^3$) corresponding to the current
57  laboratory location in Boulby mine, and one with polyhalite (K$_2$Ca$_2$Mg(SO$_4$)$_4$·2H$_2$O, 2.78 g/cm$^3$)
58  corresponding to the potential new site. Overview of the geometry can be seen in the cross-
59  sectional view of the cavern in the illustration in Fig. 1a.
60      The central part of the cryostat was the active LXe-TPC, cylindrical, 3.5 m and 2.5 m in
61  diameter and height. There was an 8 cm thick layer of LXe around the active volume separated by a
62  PTFE field cage, which will be called LXe skin hereafter. Arrays of photo-multiplier tubes (PMT)
63  meant to read the fast scintillation light from LXe and delayed electroluminescence signal from
64  Xe gas (GXe) of the TPC were approximated with uniform cylinders made of steel with reduced
65  density of 0.4 g/cm$^3$, or about 5% of the standard density of steel, simulating metal components
66  of the structure of the arrays and matching its mass. Illustration of a cross-sectional view of the
67  cryostat is shown in Fig. 1b.
68      The water tank, GdLS and LXe skin volumes were assumed to be instrumented with photo-
69  sensitive detectors and each of them would serve as a veto system by registering either Cherenkov
70  light in the case of the water tank, or scintillation light in the case of GdLS and LXe skin.

### 2.2  Underground muons

72  Cosmic-ray muons were sampled on the top and side surfaces of a 40 m cube that surrounded.
73  Distributions of primary energies and directions were calculated using the MUSIC and MUSUN
74  codes [6,7] (Ref. [7] describes the procedure and muon transport through rock down to experimen-
75  tal site). The mean muon energy and zenith angle were calculated as 261 GeV (134 GeV median)
76  and 30.6° (30.1° median), respectively. The rate of simulated muons was 0.8759 s$^{-1}$ for the exist-
77  ing Boulby site within salt at 2850 m w. e. vertical overburden. This rate was determined based on
78  the measured flux in the existing laboratory, $(3.75 \pm 0.09) \times 10^{-8}$ cm$^{-2}$s$^{-1}$ [8]. The surface profile
79  was assumed to be flat in these simulations (in reality, variations in elevation up to 30 m exist on
80  the surface over areas of a few km$^2$). For the proposed deeper site in polyhalite at 3575 m w. e.

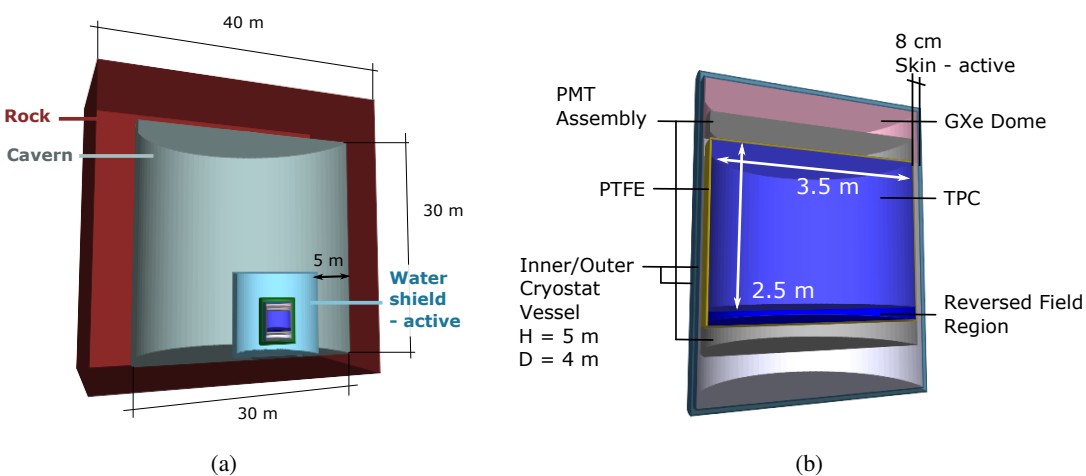

Figure 1: Illustrations of the geometry model used in the simulations. (a) Cross-sectional view of the cavern (bluish grey) within the rock (red) with the water shield (light blue) holding the scintillator (green) and the detector cryostat. (b) Cross-sectional view of the cryostat with the main LXe-TPC (blue).

vertical overburden, the same muon distributions were used, but the equivalent sampling rate was recalculated to be $0.2625 \, \text{s}^{-1}$.

We used Geant4 version 10.5 simulation toolkit to simulate the transport of muons through the modelled experimental site. Physical processes were modelled according to the toolkit's modular physics list Shielding. In total, 800 million muons were simulated for each rock material, salt and polyhalite. These numbers correspond to approximately 29 years and 97 years of live time of the experiment, respectively.

# 3   Neutron background analysis

The expected WIMP signature in a typical in a xenon-based dark matter experiment consists of a single scatter event at low energy, usually $\lesssim 50 \, \text{keV}$, which is classified as a nuclear recoil using specific discrimination techniques, and which is in anti-coincidence with veto systems. We adopted a simple background counting technique with the potential (irreducible) background identified as a single NR of energy above 1 keV.

We analysed energy depositions in active regions of the experimental setup. The detector response to the depositions was not simulated (i.e. the digitised PMT waveforms resulting from the fast scintillation light from LXe and delayed electroluminescence signal from GXe from each energy deposition in the active volume) and it was considered only in terms of the characteristic times over which signals were collected and the equivalent energy thresholds in the respective active volumes – LXe-TPC, LXe skin, liquid scintillator, water tank. We selected events with depositions above 1 keV from a single NR at least 5 cm from the boundary of the active region in 1 ms readout time window of the LXe-TPC. We required there was no other NR recoil above 0.5 keV, which we deemed resolvable in the delayed signal from GXe. We also required there were no non-NR depositions above 10 keV in total, including a quenching factor of 4 for protons and heavier ionising particles.

Energy depositions in the skin, liquid scintillator and water tank were summed over 1 μs, irrespective of their origin. We chose this time window to emulate the shaping time of the photo-sensitive instrumentation of the systems. We chose thresholds of 100 keV, 200 keV and 200 MeV

108  in the skin, liquid scintillator and water tank, respectively, to trigger veto signals. These thresh-
109  olds were inspired by our previous experience with LZ, detector modelling and the existing water
110  Cherenkov experiments, mainly the previously used energy thresholds with consideration of the
111  photo-coverage. We consider them to be conservative. The selected events in the LXe-TPC were
112  then tested for anti-coincidence with the veto signals by requiring no veto signal to be present
113  within 0.5 ms before or after any TPC signal.

114  Spectra of depositions in the LXe-TPC at various stages of the selection are shown in Fig. 2a.
115  Depositions in all events with some NR are compared to events where only NR happened. One
116  can see the low-energy part ($<100$ keV) of all the events is dominated by depositions from NR.
117  The same distributions are also shown after application of the veto condition. Recognized activity
118  near the active region of the TPC (in the LXe skin, GdLS, or the water tank) together with the
119  requirement on only NR to be observed reduces the number of events by a factor $10^6$.

120  We also wanted to answer the question whether the scintillator veto was necessary for reduc-
121  tion of the cosmogenic background. We reused the simulated samples and emulated the geometry
122  without the GdLS by assuming all depositions within its volume were part of the water tank,
123  and applied the corresponding 200 MeV threshold on the depositions from the combined volume.
124  Fig. 2b shows comparison of deposition spectra the same way is in Fig. 2a but this time the GdLS
125  was not included in the veto. The number of events in the LXe-TPC passing the veto is larger
126  than in the case with the GdLS. We must note here, that these are events before the requirement
127  on multiplicity of NR was applied.

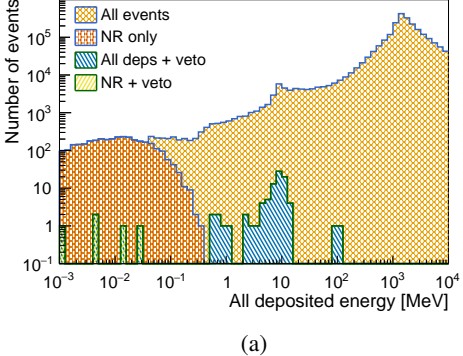
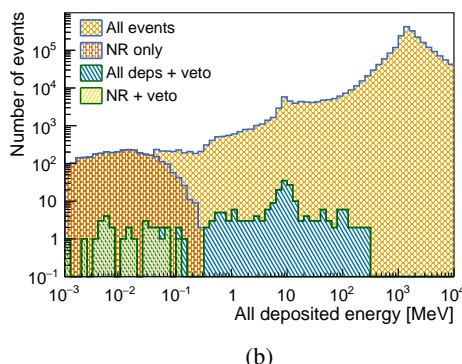

(a)                                                      (b)

Figure 2: Energy depositions in the active region of the LXe-TPC for NaCl as the rock
material. Events with any NR activity were considered. (a) Full configuration containing
the GdLS. (b) Configuration where GdLS was considered to be part of the water shield.

128  After application of the full selection process described above, including the rejection of mul-
129  tiple scatters and exclusion of 5 cm of LXe from all sides of the TPC (64 t fiducial volume), no
130  events passed the selection in the sample with NaCl as the material of the surrounding rock. A sin-
131  gle event passed the selection in the sample with polyhalite. As a result, the estimated background
132  event rate in the fiducial volume of the LXe-TPC (64 t) is $<0.84$ events (90% CL) in 10 years for
133  the 2850 m w. e. site in salt. For the 3578 m w. e. site in polyhalite the rate is estimated to be 0.10
134  (0.01–0.45 at 90% CL) events in 10 years of running.

135  In the case where no liquid scintillator is used as an additional veto system, no events were ob-
136  served for the site in salt, and the total of 2 events were observed for the site in polyhalite. The re-
137  spective estimated background rates correspond to $<0.84$ events (90% CL) in 10 years (2850 m w. e. site,
138  salt), and 0.31 (0.11–0.77 at 90% CL) events in 10 years (3575 m w. e. site, polyhalite). The limits
139  and confidence intervals are statistical only. The estimated rates in both cases are well below the
140  expected background of tens of events in 10 years extrapolated from the estimates in Table III of

Ref. [9].

The observed single nuclear recoils are located at the boundary of the fiducial volume. Most events with isolated neutrons causing NRs in the LXe-TPC and avoiding the veto were coming from delayed neutron emission from the PTFE field cage. Secondary particles in muon-induced cascade activated fluorine producing $^{17}$N from $^{19}$F. $^{17}$N undergoes $\beta$ decay with half-life of 4.2 s with emission of a neutron.

We also made considerations of systematic uncertainties. We concluded that the major uncertainty was due to the neutron production modelling in Geant4. From comparisons with previous studies (see, for instance, Refs. [8, 10] for discussions of simulations and comparison of different models with experimental data) we conservatively assessed the uncertainty to be a factor of 2. Uncertainties in the muon flux are linked to an unknown location of the new cavern within the existing level. Our estimate of 10% was based on measurements at different locations. We estimated the uncertainty of 20% for the deeper site where the flux was calculated based on the geophysical model of the Boulby mine but the exact location has not been determined. We approximated the muon spectrum at the deeper site to be the same as the one at the current laboratory. However, the calculated mean muon energies differ by about 9% (261 GeV and 282 GeV for the 2850 m w. e. and 3575 m w. e. sites, respectively) and we expect it would lead to a small increase in the neutron production yield of (6–7)%. This change is small compared with the other mentioned systematic uncertainties.

# 4   Conclusion

We have performed simulations for an experiment similar in design to the LZ detector, upscaled to 71 tonnes of LXe acting as a target. We conclude that, after applying a standard simplified analysis procedure and cuts, the event rate caused by cosmogenic activity stays below 1 event per 10 years in the fiducial volume of the LXe-TPC (64 tonnes). This rate is well below the expected background of tens of events from beta decays from radon progeny and ERs/NRs from physics backgrounds such as two-neutrino double beta decay of $^{136}$Xe and solar/atmospheric neutrinos with ER events leaking into NR band due to limited discrimination. From the point of view of cosmogenic background, the depth of about 3 km w. e. or deeper is sufficient for a next generation dark matter experiment based on liquid xenon. We have looked at simulations in different rock materials (NaCl, polyhalite, $CaCO_3$) and also with different cavern sizes and we observed that the background does not change significantly for the given detector design. Therefore we conclude that the results are applicable to other sites of similar depth to our case study of the Boulby mine.

Although there are significant systematic uncertainties related mainly to the modeling of neutron production, they cannot change our conclusion. The observed residual background of NR events comes from the production and delayed $\beta - n$ decay of $^{17}$N in PTFE (on fluorine) where only a single neutron scatter is detected. Our material budget contained about 2.8 t of PTFE. Although the residual background is very low, the design of a future experiment may need to limit PTFE usage to the necessary minimum.

We have also investigated two designs of veto system: a default one with instrumented liquid scintillator surrounding the cryostat, and an option without the scintillator. No significant difference was observed between the two scenarios which lead us to the conclusion that the additional veto system is not required to suppress cosmogenic backgrounds for the sensitivity at the studied depth. This conclusion, however, does not apply to other types of backgrounds (mainly from detector components) where liquid scintillator is particularly efficient in tagging neutron-induced events. The decision about the need for a scintillator veto can only be taken after a detailed simulation of all types of background from all components is completed.

## Acknowledgments

We wish to thank UKRI-STFC for financial support of the Boulby case study and the whole team who worked on the Boulby Feasibility Study project.

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
