# Peer review of "Muon-induced background in a next-generation dark matter experiment based on liquid xenon"

_SciPost Physics Proceedings_

## Round 1 · Referee Report · Anonymous (Referee 1) · 2022-11-8

Strengths

Very valuable work on the simulation of muon-induced background at two depths (existing and planned) for next generation Liquid Xenon TPC experiments. Even with rather high uncertainties the result is reliable and shows comparable outcome at both depth.

Weaknesses

some additional formatting needed. Can be done in the stage of post production.

Report

The manuscript is clearly written and well organized. It is also suitably formatted for publication.
I recommend the manuscript for publication after some changes.

Requested changes

  • in the abstract I suggest to make clear about which experiment you are talking.
  • neutrino floor -> fog?
  • you simulate two types rock composition, but say that results are comparable for other labs. I would add explanation why.
  • double introduction of NR
  • delete comment before abstract (in red)
  • I would suggest authors explain why given threshold was chosen for more clarity of the paper.

  • validity: -
  • significance: top
  • originality: high
  • clarity: high
  • formatting: excellent
  • grammar: excellent

Author:  Viktor Pěč  on 2022-11-14  [id 3016]

(in reply to Report 1 on 2022-11-08)

Comment to requested change 1:
We prefer not to name any specific potential experiment as the study was aimed to be more generic. We belief that the important qualities of the simplified design model are stated in the abstract. The motivation for the chosen design is described in section 2.1.

We have made changes to the manuscript correspondingly according to the other comments/suggestions.

---

## Round 2 · Author Response

We have read through the report and changed the manuscript accordingly, see the list of changes.

We did not act upon the requested change 1. We prefer not to name any specific potential experiment as the study was aimed to be more generic. We belief that the important qualities of the simplified design model are stated in the abstract. The motivation for the chosen design is described in section 2.1.

---

## Round 2 · List of Changes

i. added "(fog)" after 2 mentions of "neutrino floor" in the abstract. (Re. request 2)

ii. added 2 sentences in the Conclusion section (Re. request 3):
"We have looked at simulations in different rock materials (NaCl, polyhalite, CaCO$_3$) and also with different cavern sizes and we observed that the background does not change significantly for the given detector design. Therefore we conclude that the results are applicable to other sites of similar depth to our case study of the Boulby mine."
Right after
"From the point of view of cosmogenic background, the depth of about 3\,k\mwe\ or deeper is sufficient for a next generation dark matter experiment based on liquid xenon."

iii. removed "(NR)" in section 3. (Re. request 4)

iv. deleted text in red on the front page (Re. request 5)

v. adde a sentence in section 3 (Re. request 6) after
"We chose thresholds of 100\,keV, 200\,keV and 200\,MeV in the skin, liquid scintillator and water tank, respectively, to trigger veto signals.":
"These thresholds were inspired by our previous experience with LZ, detector modelling and the existing water Cherenkov experiments, mainly the previously used energy thresholds with consideration of the photo-coverage."

You are currently on this page

Resubmission scipost_202210_00041v2 on 14 November 2022

---

## Editorial Decision

accepted_in_target_journal